# Fine-Scale Sea Ice Segmentation for High-Resolution Satellite Imagery with Weakly-Supervised CNNs

**Bento C. Gonçalves** [1,2,*] **and Heather J. Lynch** [1,2]

1 Department of Ecology & Evolution, Stony Brook University, Stony Brook, NY 11790, USA;
heather.lynch@stonybrook.edu
2 Institute for Advanced Computational Science, Stony Brook University, Stony Brook, NY 11794, USA
* Correspondence: bento.goncalves@stonybrook.edu

**Abstract:** Fine-scale sea ice conditions are key to our efforts to understand and model climate change. We propose the first deep learning pipeline to extract fine-scale sea ice layers from high-resolution satellite imagery (Worldview-3). Extracting sea ice from imagery is often challenging due to the potentially complex texture from older ice floes (i.e., floating chunks of sea ice) and surrounding slush ice, making ice floes less distinctive from the surrounding water. We propose a pipeline using a U-Net variant with a Resnet encoder to retrieve ice floe pixel masks from very-high-resolution multispectral satellite imagery. Even with a modest-sized hand-labeled training set and the most basic hyperparameter choices, our CNN-based approach attains an out-of-sample F1 score of 0.698–a nearly 60% improvement when compared to a watershed segmentation baseline. We then supplement our training set with a much larger sample of images weak-labeled by a watershed segmentation algorithm. To ensure watershed derived pack-ice masks were a good representation of the underlying images, we created a synthetic version for each weak-labeled image, where areas outside the mask are replaced by open water scenery. Adding our synthetic image dataset, obtained at minimal effort when compared with hand-labeling, further improves the out-of-sample F1 score to 0.734. Finally, we use an ensemble of four test metrics and evaluated after mosaicing outputs for entire scenes to mimic production setting during model selection, reaching an out-of-sample F1 score of 0.753. Our fully-automated pipeline is capable of detecting, monitoring, and segmenting ice floes at a very fine level of detail, and provides a roadmap for other use-cases where partial results can be obtained with threshold-based methods but a context-robust segmentation pipeline is desired.

**Keywords:** pack-ice; worldview 3; semantic segmentation; deep learning; remote sensing image processing

## 1. Introduction

Antarctic sea ice is an exceptionally dynamic habitat that plays an important role in climate feedback cycles [1,2] and controls either directly or indirectly the Southern Ocean food web [3–5]. While coarse-grained maps of Antarctic sea ice have been available for several decades [6,7], and have been critical to safe navigation [8,9], climate modelling [10] and our understanding of sea ice-dependent predators [11], current sea ice products are primarily derived from passive microwave sensors operating at 25 km resolution and are therefore too coarse to resolve individual floes. Moreover, marine predators such as penguins and seals interact with sea ice on an extremely localized basis, and the characteristics of sea ice that might influence decisions about movement, foraging, or reproduction occur at scales far smaller than the resolution of typical sea ice imagery products [12–14]. Sub-meter resolution satellite imagery is now widely available for the Antarctic and this provides an opportunity to start mapping sea ice conditions over large spatial scales. The development and availability of fine-scale sea ice data products will radically expand our capacity to create high-resolution sea ice charts for navigation, link

observed fine scale sea ice conditions to climate models, and to understand the detailed habitat requirements of sea ice dependent predators. Mapping fine-scale sea ice conditions at scale, especially within the highly-heterogeneous pack-ice zone, will require automated pipelines for sea ice segmentation.

Sea ice extraction is an active field in remote sensing. Typically, sea ice layers are extracted from Synthetic Aperture Radar (SAR) and optical sensors of low to medium resolution (e.g., MODIS, Sentinel-2, Landsat). Traditionally, sea ice was identified using pixel-based methods that used only the information contained in the spectral profile of each individual pixel to extract sea ice masks [15]. Other approaches explored the contrast between sea ice and the surrounding water bodies and threshold (e.g., watershed segmentation) or clustering based methods (e.g., k-means clustering) to extract sea ice polygons [16–18], without the need of labeled datasets. More recently, machine learning models were trained to identify and predict different sea ice types from predetermined sets of sea ice polygons and expert-annotated class labels (e.g., [19–21]). Approaches using low to medium resolution sensors bring the advantages of larger spatial and temporal coverage and, in the case of passive microwave and other non-optical sensors, the capability to extract useful information regardless of cloud cover and other factors that affect lighting. Although such methods have provided invaluable information on traits such as average sea ice cover, they are unfit to extract individual ice floes or fine-grained information on sea ice conditions. In imagery from very-high-resolution sensors such as Worldview-3, individual ice floes are several pixels large, and the classification and delineation of such super-pixel features is highly challenging for pixel-based solutions. Moreover, the extra detail adds a larger breadth of features that can hinder the performance of threshold based methods. Fortunately, modern computer vision (CV) approaches exploiting deep learning (DL) are well suited to exactly such problems.

The rise of GPU-accelerated DL, marked by the first Imagenet challenge won by a Convolutional Neural Network (CNN) [22], has made DL affordable, brought the field back as a hot research topic and ultimately lead several ground-breaking improvements to the fields of CV and natural language processing (NLP). With the concomitant popularization of high-resolution sensors, DL solutions have largely replaced methods such as Support Vector Machines (SVM) and has already become a staple in some areas of remote sensing [23]. In contrast to other works that use DL for classifying sea ice at medium resolution (e.g., [20,21]) and segment out sea ice in ship-borne images [24,25], the goal of the present work is extracting precise ice floe masks from high-resolution imagery. More specifically, we are targeting ice floes only–a daunting task given the large number of potentially confounding fine-scale structures (e.g., slush, melt ponds, etc.) that emerge at very-high spatial resolutions. We do so by training a weakly-supervised CNN that learns from a small set of hand-labeled sea ice masks and a much larger set of weak annotations obtained with minimal effort using a watershed segmentation model. A fully automated pack-ice extraction tool would provide invaluable data for Antarctic ecology given the large number of ecosystem interactions mediated by sea ice.

## 2. Materials and Methods

### 2.1. Imagery and Data Annotation

Our datasets were extracted from a set of 43 multispectral Worldview-3 scenes (Table 1 and Figure 1) covering 730.05 km$^2$ of coastal Antarctic scenery with an on-nadir resolution of 1.24 m/pixel. We include three distinct types of annotation (Table 2): (1) pixel level sea ice masks drawn by hand–our "hand-labeled" training set; (2) pixel level sea ice masks extracted with watershed segmentation–our "watershed" training set; and (3) pixel level sea ice masks extracted with watershed segmentation and adapted to synthetic sea ice images–our "synthetic" training set. This multi-dataset design allows us to take advantage of weak labels from watershed segmentation (watershed and synthetic training sets) during training but still get validation and test metrics on a set of careful manual annotations. Each scene consisted of the red, green and blue bands of the WV-3 multispectral image tiled

into 784 × 784 pixel patches with a 50% overlap between neighboring patches. We chose to extract patches that are bigger than our input size to generate a larger breadth of training images by leveraging random-crops within our data-augmentation pipeline (described in the following section). Details on each method are supplied in the following sections.

**Table 1.** WorldView-3 imagery. We used a set of 43 multispectral WV-3 images to train, validate and test our ice floe segmentation models. To reduce GPU memory footprint during training and avoid further modifications to our CNN architectures, all imagery was converted from the native 8-band multispectral channels to three channel images by extracting the red, green and blue bands. Due to lighting limitations inherent to the poles and to capture the reproductive seasons of Antarctic megafauna, our imagery was acquired in a period ranging from November 20 to April 7 (summer-early spring) in the years of 2014–2017. All the imagery used in the study is cloud-free. Repeated consecutive catalog IDs indicate different scenes within the same strip.

| Catalog ID | Lat-Lon | Cloud Cover | Total Area | Date |
|---|---|---|---|---|
| 1040010005B62F00 | −69.3327 158.4884 | 0.0 | 263.1 km$^2$ | 20 November 2014 |
| 104001001334670 | −76.9427 166.8715 | 0.0 | 212.6 km$^2$ | 26 November 2015 |
| 10400100156E6500 | −63.1618 −54.9593 | 0.0 | 268.8 km$^2$ | 01 January 2016 |
| 10400100156E6500 | −63.8006 −54.959 | 0.0 | 202.7 km$^2$ | 01 January 2016 |
| 10400100156E6500 | −63.2718 −54.959 | 0.0 | 265.3 km$^2$ | 01 January 2016 |
| 10400100156E6500 | −63.599 −54.9589 | 0.0 | 259.3 km$^2$ | 01 January 2016 |
| 1040010016234E00 | −67.256 45.9485 | 0.0 | 266.5 km$^2$ | 02 January 2016 |
| 1040010016234E00 | −67.668 45.9477 | 0.0 | 172.8 km$^2$ | 02 January 2016 |
| 1040010016234E00 | −67.0437 45.9485 | 0.0 | 244.6 km$^2$ | 02 January 2016 |
| 1040010016234E00 | −67.1471 45.9486 | 0.0 | 265.0 km$^2$ | 02 January 2016 |
| 1040010016234E00 | −67.3652 45.9489 | 0.0 | 268.2 km$^2$ | 02 January 2016 |
| 1040010016234E00 | −67.4748 45.9489 | 0.0 | 269.9 km$^2$ | 02 January 2016 |
| 1040010017265B00 | −76.0 −26.6717 | 0.0 | 224.5 km$^2$ | 07 January 2016 |
| 1040010017A12200 | −67.4771 164.6313 | 0.0 | 168.7 km$^2$ | 12 January 2016 |
| 10400100167EC800 | −63.4564 −56.8695 | 0.0 | 282.7 km$^2$ | 17 January 2016 |
| 10400100167EC800 | −63.3475 −56.8686 | 0.0 | 281.0 km$^2$ | 17 January 2016 |
| 10400100167EC800 | −63.6757 −56.8695 | 0.0 | 287.3 km$^2$ | 17 January 2016 |
| 10400100167EC800 | −63.2385 −56.8685 | 0.0 | 279.2 km$^2$ | 17 January 2016 |
| 10400100178F7100 | −63.4235 −54.669 | 0.0 | 186.1 km$^2$ | 21 January 2016 |
| 104001001762AC00 | −66.2365 110.1896 | 0.0 | 191.1 km$^2$ | 21 January 2016 |
| 10400100175A5600 | −66.6168 −68.2485 | 0.0 | 122.0 km$^2$ | 25 January 2016 |
| 10400100175A5600 | −67.575 −68.25 | 0.0 | 269.3 km$^2$ | 25 January 2016 |
| 104001001747E000 | −64.2565 −56.6693 | 0.0 | 291.3 km$^2$ | 26 January 2016 |
| 1040010017777C600 | −69.0697 76.7836 | 0.0 | 220.4 km$^2$ | 28 January 2016 |
| 1040010018447F00 | −67.6175 66.5771 | 0.0 | 296.5 km$^2$ | 28 January 2016 |
| 104001001844A900 | −66.5325 92.5386 | 0.0 | 208.0 km$^2$ | 28 January 2016 |
| 1040010017764300 | −74.7749 164.0267 | 0.0 | 225.3 km$^2$ | 29 January 2016 |
| 1040010017823400 | −72.3657 170.2705 | 0.0 | 207.9 km$^2$ | 04 February 2016 |
| 1040010018694800 | −72.0 170.5882 | 0.0 | 170.7 km$^2$ | 04 February 2016 |
| 10400100196BE200 | −65.4111 −64.3911 | 0.0 | 274.8 km$^2$ | 25 February 2016 |
| 10400100196BE200 | −65.4984 −64.3908 | 0.0 | 191.9 km$^2$ | 25 February 2016 |
| 10400100181F9B00 | −66.8013 50.5412 | 0.0 | 215.6 km$^2$ | 27 February 2016 |
| 1040010018755100 | −67.4705 61.0185 | 0.0 | 221.4 km$^2$ | 05 March 2016 |
| 1040010018046800 | −65.938 110.2305 | 0.0 | 207.7 km$^2$ | 07 March 2016 |
| 1040010019529D00 | −77.7016 −47.6769 | 0.0 | 183.9 km$^2$ | 13 March 2016 |
| 1040010019417700 | −76.1377 168.3823 | 0.0 | 243.9 km$^2$ | 15 March 2016 |
| 104001001A625A00 | −70.0097 −1.4187 | 0.0 | 163.3 km$^2$ | 16 March 2016 |
| 104001001A8FF900 | −67.3803 63.9762 | 0.0 | 237.1 km$^2$ | 16 March 2016 |
| 104001001A27CC00 | −64.5113 −57.4442 | 0.0 | 264.6 km$^2$ | 23 March 2016 |
| 104001001B448400 | −69.9403 8.3095 | 0.0 | 163.1 km$^2$ | 25 March 2016 |
| 104001001A896700 | −67.8698 69.7022 | 0.0 | 181.1 km$^2$ | 30 March 2016 |
| 104001001A6C8C00 | −70.5887 −60.5685 | 0.0 | 234.1 km$^2$ | 07 April 2016 |
| 1040010028CD9C00 | −73.2326 −126.7786 | 0.0 | 162.3 km$^2$ | 25 January 2017 |

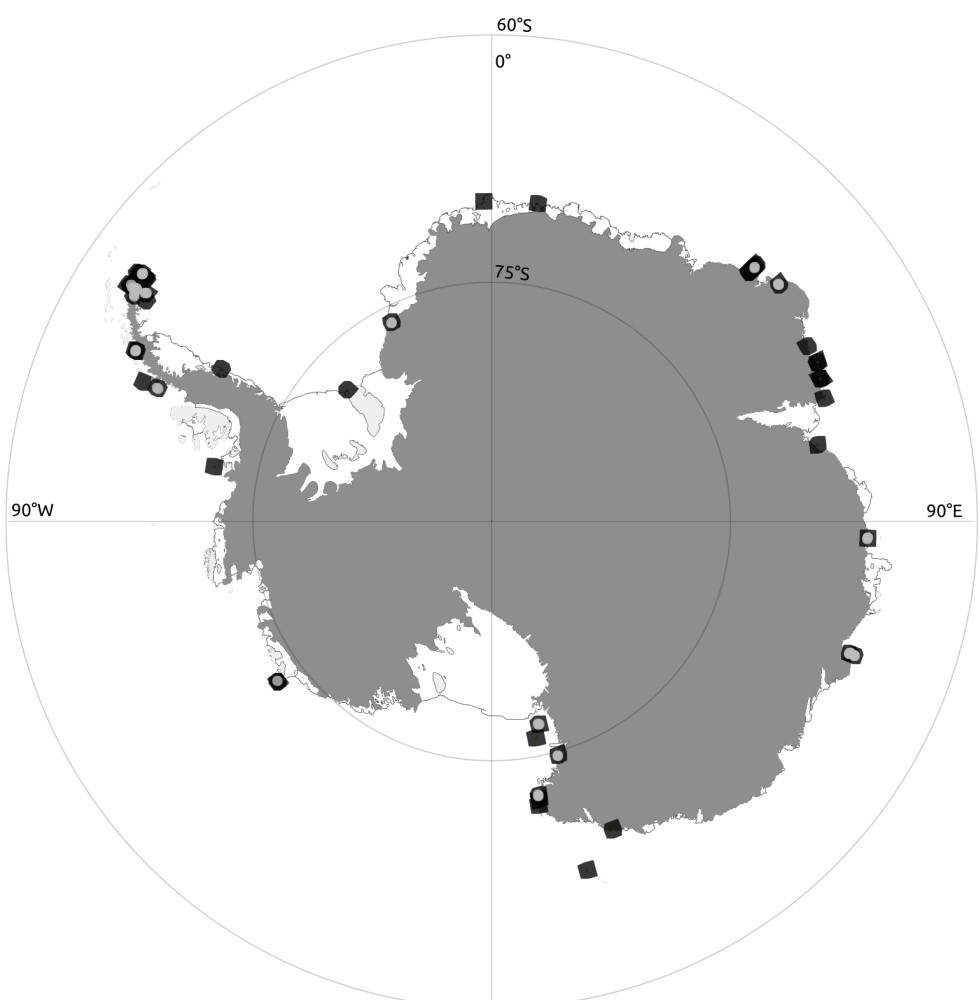

**Figure 1.** Training set scenes. Dark squares denote the location of each of the 36 scenes in our training set. Scene squares are marked with a light dot whenever we drew annotations by hand for that specific scene. Imagery copyright Maxar, Inc., Westminster, CO, USA, 2021.

**Table 2.** Training datasets. Number of scenes and total area covered by positive (i.e., patches with pack-ice) and negative (i.e., patches without pack-ice) patches within each of our datasets. Image annotations consisted of binary pixel masks that denote whether a pixel in a patch represents pack-ice. Number of scenes and areas covered by each of our training sets. Hand-labeled masks were drawn over 3000 × 3000 m crops at strategic locations whereas watershed derived masks were extracted by running a sliding window over scene regions marked by irregular polygons. Patches with watershed derived masks are exclusively used during training, whereas patches with hand-labeled masks are split equally between training, validation and test sets. Negative training patches were shared across all three training sets. To avoid inflation in our validation metric scores, we set aside a distinct set of negative images for validation.

| Training Set | Scenes | Area + | Area − |
|:---:|:---:|:---:|:---:|
| Hand-labeled [train] | 19 | 20.8 km$^2$ | 240.9 km$^2$ |
| Hand-labeled [valid] | 18 | 20.2 km$^2$ | 17.85 km$^2$ |
| Hand-labeled [test] | 19 | 20.4 km$^2$ | 16.8 km$^2$ |
| Watershed [train] | 27 | 393.1 km$^2$ | 240.9 km$^2$ |
| Synthetic [train] | 27 | 393.1 km$^2$ | 240.9 km$^2$ |

### 2.1.1. Hand-Labeled Training Set

We employed hand-labeled pixel masks as our main tool to provide out-of-sample performance measurements to segmentation CNNs. Our hand labeled masks were created by following steps: (1) extracting three RGB 3000 × 3000 pixel crops containing pack-ice at random, but with no overlap, from 5 different scenes; (2) opening the crops in Adobe Photoshop™ and creating a separate channel to store our sea ice mask; (3) using the magic wand and color selection tools to remove darker regions containing open water from our sea ice mask; and (4) manually filling holes created by darker areas inside floes. All our manual annotations were performed by a single individual and included multiple passes over the dataset to guarantee that annotations were as consistent as possible across different scenes. Our crops and pixel masks were tiled using a sliding window approach with a patch-size of 784 × 784 and 50% overlap between neighboring patches. We further supplemented this dataset by adding hard-negative patches (i.e., without sea ice) at the same proportion as the following two datasets. The final hand-labeled dataset is drawn from 45 hand-labeled RGB crops split equally between training, validation and test sets.

### 2.1.2. Watershed Training Set

We used a watershed segmentation algorithm as an inexpensive strategy to generate a large number of weak ground-truth masks from raw imagery with sea ice, as follows: (1) create georeferenced annotation masks by hand-drawing contour polygons over areas with pack-ice; (2) mask raw imagery and run a sliding window with a patch size of 784 and 50% overlap between neighboring patches to extract input patches; (3) Create an annotation mask for each patch by running watershed segmentation sequentially; (4) draw contours for objects in the watershed mask and remove objects that are deemed too small to be a floe from the watershed mask (total area < 15 m$^2$; (5) discard images where more than 15% of the total area has missing data or the watershed mask has a single contour (usually a contrast aberration around corners). We added extra patches in an equivalent manner using georeferenced polygons drawn in representative areas outside of pack-ice to serve as hard-negatives. The final Watershed training set contains a total 6597 patches divided into 4085 pack-ice images and 2512 hard-negative patches.

### 2.1.3. Synthetic Image Training Set

We built upon the previous dataset by creating synthetic versions of the imagery where the input image better matches its watershed mask as follows (Figure 2): (1) taking a patch with pack-ice; (2) applying recursive watershed segmentation to the patch; (3) using the output of watershed segmentation to mask out all portions of the patch that did not contain sea ice; (4) pasting the resulting patch on top of an open water background patch to create a realistic synthetic image; and (5) removing the areas of greatest overlap between masked RGB channels from the segmentation mask to further individualize floes in the mask. The final Synthetic dataset has the same number of patches as the original Watershed dataset.

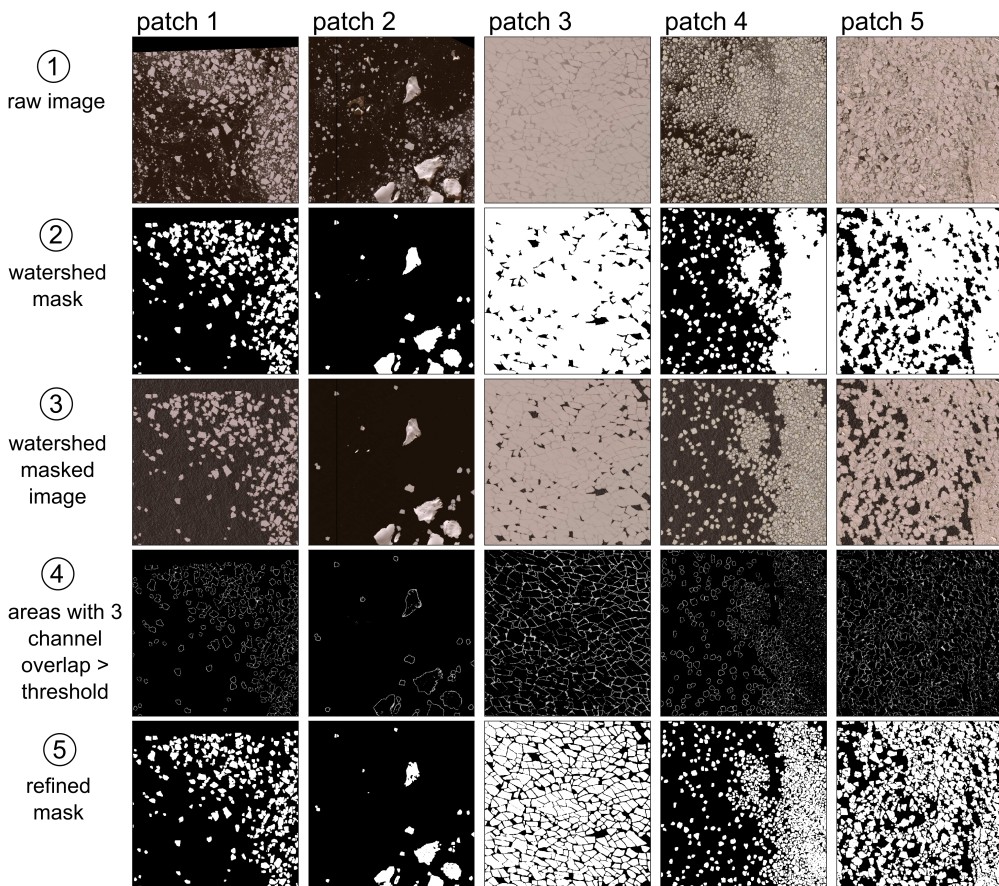

**Figure 2.** Synthetic image creation. Five examples of our synthetic image creation pipeline extracted from our training set images. Our watershed segmentation algorithm in step 2 is applied sequentially for a total of three times. We fill masked-out areas in step 3 with open water images sampled at random. We find three channel overlaps in step 4 using an adaptive threshold. Refined masks in step 5 are obtained by subtracting overlapping areas from step 4 from watershed masks in step 2. Imagery copyright Maxar, Inc., Westminster, CO, USA, 2021.

### 2.2. Segmentation CNNs

We use a U-Net variant [26] with a ResNet34 encoder [27] as the down sampling branch of our CNN architecture for segmentation (Figure 3), trained to create pixel-level binary masks that represent which areas of a patch are covered by pack-ice. We make this small modification to make our encoder branch more flexible because of the skip-connections within ResNet convolution blocks and easily allow experimenting with fine-tuning from a ResNet classifier. We used Dice coefficient as our validation metric for model selection. We kept the best performing model for each training set for comparison against the hand-labeled test dataset to get an out-of-sample measurement of model performance, boosted by test-time-augmentation [28]. Finally, we took the best performing model according to test F1 score and retrained it on all samples from the synthetic and hand-labeled dataset to be used in production.

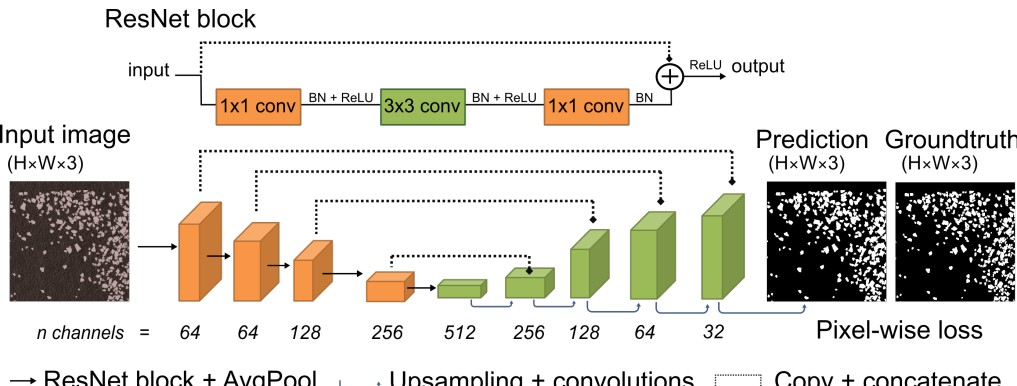

**Figure 3.** CNN architecture. Our CNN architecture borrows from the U-Net architecture, with encoder and decoder branches connected by copy-and-concatenate operations, with the sole difference that the base U-Net encoder is replaced with a ResNet34 encoder. ResNet blocks within the encoder consist of a set of convolution operators intertwined by batch normalization and rectified linear unit (ReLU) operations followed by a concatenation with the input features (i.e., skip-connection). After running through ResNet blocks, features get down-sampled after each ResNet block with a strided Average pooling layer, reducing the height and width of each channel by a factor of 2. We do not provide numbers for height and width for input images and CNN blocks in the schematic because input size is a dynamic parameter in our study design.

### 2.3. CNN Training and Validation

All our CNN training experiments were run on PyTorch v1.8.0 in Python [29], with an Adam optimizer [30] and a schedule where the learning rate is reduced by factor of 10 whenever the validation F1 score fails to improve after three consecutive epochs and training is interrupted after six epochs without improving validation F1. We searched for optimal combinations of hyperparameters running over 1000 random search experiments for input size (256, 384 and 512), loss function (see below), data augmentation (simple vs. complex, see below) and learning rate (log scale, from $1 \times 10^{-3}$ to $1 \times 10^{-5}$), using the greatest batch size allowed by GPU memory (200, 120 and 60 for input sizes 256, 384 and 512, respectively) and validation F1 score as the model selection metric. Our training images are sampled with replacement to match a predefined ratio of negative to positive images in training batches, which was also explored as a hyperparameter. To explore the benefits of fine-tuning from a pre-trained model [31], we repeated our hyperparameter search experiments initializing model parameter weights to either the weights from one our best performing models according to validation metrics (picked at random from the top 100 models) or a CNN trained on binary classification for the presence of pack-ice in patches. Our experiments were run at the Bridges-2 NSF supercomputer on GPU nodes with eight Nvidia V-100 GPUs, each with 32 GB of GPU memory. Model weights for our best-performing segmentation models are available in our GitHub code repository. We grouped our random search experiments with two options (i.e., data augmentation policy, fine-tuning and test-time-augmentation), within 12 brackets defined by combinations of input size and training set and treated them as independent replicates for statistical analyses. More specifically, we extracted the best performing model within each of our 12 brackets and tested whether the observed ratio of best performing models given parameter values falls within the expectations of a binomial experiment with 12 trials and 50% probability of success.

### 2.4. Testing

We tested our CNNs using a routine that mimics the functioning of our models in production consisting of four steps: (1) use a sliding window approach to tile the input image into patches with a 50% overlap between neighboring patches, where the size of each patch matches the required input size of the CNN (i.e., 256, 384 or 512); (2) generate

predictions for each patch by applying a sigmoid transformation and binary threshold to the model output; and (3) create a mosaic of the predicted tiles and calculate metric scores by comparing predictions with ground-truth masks. Each of our CNN models from training were tested with and without test-time-augmentation, using a temperature-sharpen policy [32] to merge augmented predictions. To get robust out-of-sample performance estimates, our model selection used a consensus of four different metrics: (1) mean F1 score averaged across test scenes; (2) mean IoU averaged across test scenes; (3) F1 score across all pixels in the test set; and (4) average between the accuracy on background and foreground pixels in the test set. For each candidate model, we ran this pipeline over a set of 12 carefully labeled 3000 × 3000 m areas with pack-ice and seven 3000 × 3000 m areas of Antarctic scenery without pack-ice.

### 2.5. Loss Functions

We experimented with a variety of loss functions that focus on different aspects of the segmentation output, largely borrowing from a recent comprehensive survey on loss functions for semantic segmentation CNNs [33]. Since the choice of loss function can have dramatic, non-obvious impacts in model performance [33], we chose to start with a broad set of candidate loss functions and use validation F1 scores during the hyperparameter search to find the ideal candidate for our use case. We initially used two pixel-based approaches, namely binary cross-entropy and Focal loss [34]. While the former represents the simplest available loss function and is ideal for a baseline, the latter is often used for imbalanced datasets, as it puts more weight into pixels that are harder to classify. We then tested a number of region-based approaches that build upon the Dice coefficient [35] as they tend to preserve the shape of superpixel structures better than pixel-based solutions. In the context of semantic segmentation models, the Dice coefficient, a harmonic mean of precision and recall, is turned into a loss function by subtracting the Dice coefficient for a patch from 1, so that models can improve by minimizing it through gradient descent optimization. Besides the original Dice Loss, we used three variants: (1) Log-Cosh Dice loss [33], an attempt to improve the original Dice loss by smoothing out its loss function; (2) Dice Perimeter loss [36], a variation of Dice loss that uses the difference in the total perimeter of the predicted and ground truth masks as a regularization factor to the loss function; and (3) a weighted mixture of Dice and Focal Loss. Whenever available, we used native PyTorch implementations of our loss functions.

### 2.6. Data Augmentation

To add more breadth to our training sets, and consequently make make our model more robust to changes in scale, rotation, illumination and position, we employed data augmentation pipelines tailored for satellite imagery, taking full advantage of rotations and random crops that would otherwise be unsuitable for non-aerial images. We use two data augmentation strategies: (1) a simple approach with random-crops, vertical and horizontal flips, random shifts in position, random re-scaling, random 90-degree rotations (i.e., 90, 180, or 270 degrees), and brightness and contrast shifts; and (2) a more complex approach using the same transforms listed above plus noise reduction, RGB shifts, and random distortion effects. Our data augmentation pipelines are applied continuously during training and use transforms implementations from the Albumentations package [37]. The exact specifications for each can be found in our GitHub code repository (accessed on 15 August 2012).

### 2.7. Model Baselines

We evaluated our sea ice extraction models using 4 baselines of increasing complexity: (1) watershed: extract directly with watershed segmentation (identical implementation from our watershed training set extraction); (2) basic U-Net: use the best performing U-Net constrained to the simplest settings in our hyperparameter search (hand-labeled training set, binary cross-entropy loss, simple augmentation pipeline, no model fine-tuning, no

test-time augmentation, validation F1 score as model selection); (3) U-Net best validation: the best performing U-Net from the hyperparameter search according to validation F1 scores; and (4) U-Net best test: the best performing U-Net according to an ensemble of 4 different metrics measured after mosaicing model output. We obtained an out-of-sample performance estimate for each baseline as described in Section 2.4.

## 3. Results

### 3.1. Model Performance

Our first baseline, applying watershed segmentation to input images, attains a 0.464 F1 score in the test set after output mosaicing (Table 3). The simplest possible CNN-based model improves performance by >50%, reaching a test F1 score of 0.698. Adding more complex features and the synthetic dataset to the hyperparameter search (Figure 4) model, obtained by adopting a more elaborate model selection approach that mimics production settings, provides another modest improvement in terms of test F1 score, reaching 0.753.

**Table 3.** Model performance. We show the F1 scores on validation and test sets of the best performing model iteration across brackets input size and dataset as well as the number of random search experiment runs within each bracket trained from randomly initialized parameter weights (i.e., from scratch) or fine-tuning from a previous model, respectively. Validation F1 scores are obtained by averaging out the F1 scores from individual patches in the validation set. Test F1 scores reported are averages across the F1 score for all 19 test scenes obtained after output patches were merged into a mosaic, more akin to production settings, with the standard error as a measurement of spread. Test F1 scores from the same watershed segmentation approach we used to extract weakly-labeled images are provided as a baseline for U-Net based models. Our watershed segmentation model is implemented in Python using the numpy and OpenCV libraries and our U-Net CNN is implemented in PyTorch by swapping the original U-Net down-sampling layer for a ResNet34 encoder.

| Model | Input Size | Dataset | F1 (Val) | F1 (Test) | N |
|---|---|---|---|---|---|
| U-Net | 256 | hand | 0.842 | $0.727 \pm 0.132$ | 34, 12 |
| U-Net | 256 | hand + synthetic | 0.824 | $0.713 \pm 0.87$ | 36, 16 |
| U-Net | 256 | hand + watershed | 0.855 | $0.628 \pm 0.174$ | 34, 12 |
| U-Net | 256 | synthetic | 0.732 | $0.739 \pm 0.126$ | 42, 17 |
| Watershed | 256 | - | - | $0.464 \pm 0.139$ | - |
| U-Net | 384 | hand | 0.736 | $0.747 \pm 0.142$ | 31, 16 |
| U-Net | 384 | hand + synthetic | 0.822 | $0.713 \pm 0.162$ | 41, 19 |
| U-Net | 384 | hand + watershed | 0.848 | $0.633 \pm 0.180$ | 33, 10 |
| U-Net | 384 | synthetic | 0.769 | $0.727 \pm 0.135$ | 46, 21 |
| Watershed | 384 | - | - | $0.460 \pm 0.141$ | - |
| U-Net | 512 | hand | 0.776 | $0.733 \pm 0.158$ | 40, 13 |
| U-Net | 512 | hand + synthetic | 0.850 | $0.753 \pm 0.113$ | 32, 14 |
| U-Net | 512 | hand + watershed | 0.839 | $0.696 \pm 0.176$ | 39, 14 |
| U-Net | 512 | synthetic | 0.830 | $0.734 \pm 0.133$ | 37, 14 |
| Watershed | 512 | - | - | $0.459 \pm 0.136$ | - |

### 3.2. Hyperparameter Search

Our hyperparameter search experiments (Figure 4) unanimously favored the use of test-time-augmentation ($\rho = 0.00024$, best performance in 12 out of 12 brackets), and showed a slight, non-significant support for the use of our simple data augmentation pipeline over the complex one ($\rho = 0.07299$, best performance in 9 out of 12 brackets) and training from scratch instead of fine-tuning ($\rho = 0.07299$, best performance in 9 out of 12 brackets). In general, F1 score differences between different parameter choices were much smaller when fine-tuning from previous models.

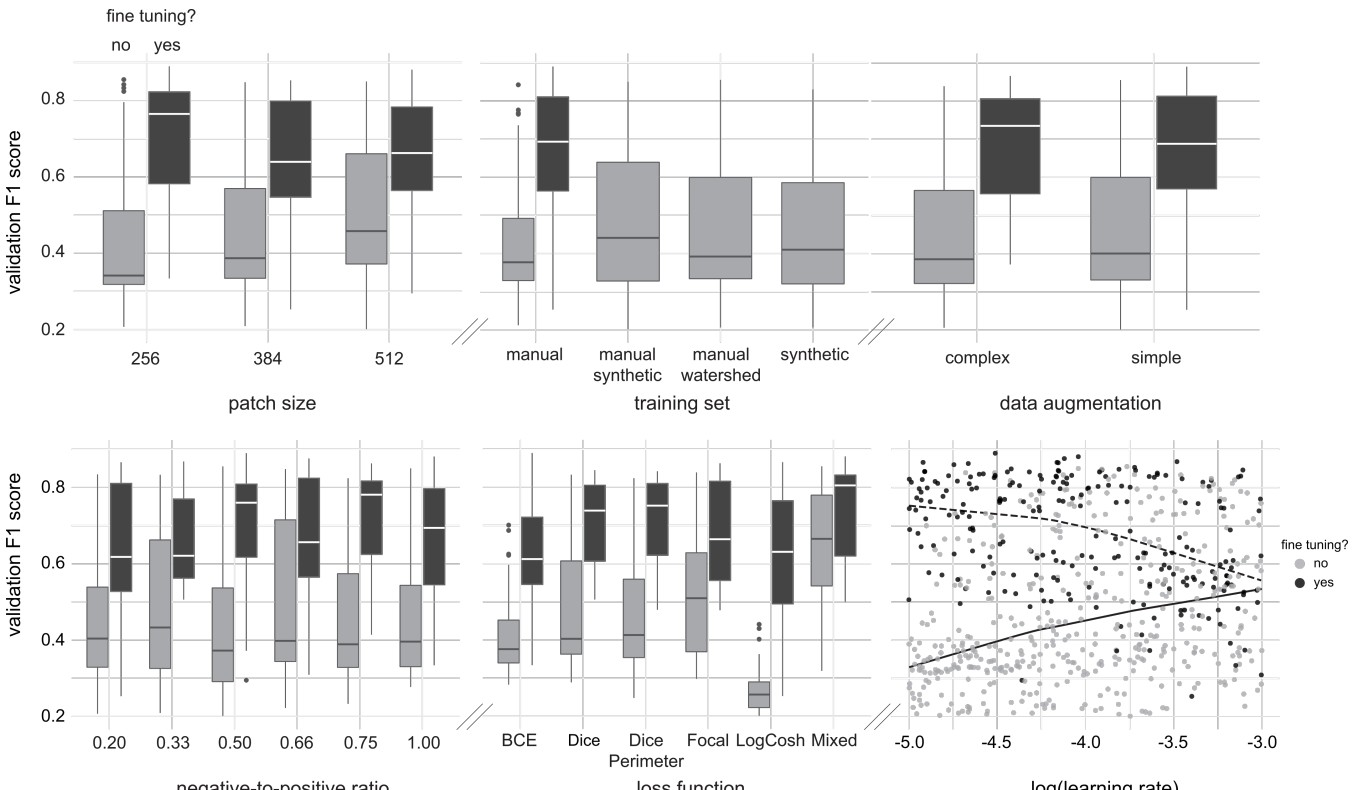

**Figure 4.** Hyperparameter tuning experiments. Validation F1 scores of 623 random search experiments across six different hyperparameters. We test the influence of input size, training set choice, data augmentation routine, ratio of negative to positive images within mini-batches, choice of loss function and learning rate on model performance, measured as the F1 score in the validation set. Our training sets consist of combinations of a small set of hand-labeled images ("manual"), a larger set of images annotated using a watershed segmentation algorithm ("watershed") and a set of synthetic input images created by modifying images from the previous set to be more consistent with their watershed-derived masks ("synthetic"). For loss functions, we tested binary cross-entropy loss (BCE), Focal Loss, three variants of Dice loss, and a weighted mixture of Dice and Focal losses. For each experiment, we split our runs between models trained from scratch and models fine-tuned from a previous experiments, in which case initial parameter weights would be drawn from one of the top 100 models trained from scratch, selected at random. All our fine-tuning experiments were trained with manual labels, as the annotation masks within are closer to the output than we would wish during inference. The learning rate scatter plot shows each experiment as a dot and trend lines for models trained from scratch (continuous line) and fine-tuned models (dashed line).

### 3.3. Qualitative Model Output

Model predictions obtained with watershed segmentation produce several false-positive and false-negative errors in scenes with pack-ice and produces an abundance of false-positive errors in background scenes (Figure 5). Our most basic CNN model has a greater recall than the previous baseline, at the cost of a lower precision in the third pack-ice scene, and successfully discards some icebergs and rocks from the predicted mask. Though it incurs substantially fewer false-positive errors than the previous baseline in background scenes, it does generate artifacts around edges for those. The best model according to validation metrics produces sharp prediction masks inside pack-ice scenes but largely fails to discard icebergs and rocks from the predicted mask. Though this baseline achieves a higher overall F1 score than the previous one, it largely fails to ignore background imagery, incurring substantial false-positive errors at those. Our final model, picked by our enhanced model selection scheme, has a lower recall but higher precision in pack-ice scenes when compared to the previous baseline and consistently discards icebergs and rocks from the predicted mask. Unlike the other three baselines, our final model generates little to no

false-positives when predicting outside of pack-ice (7 out of 7 background scenes had less than 0.5% false-positives).

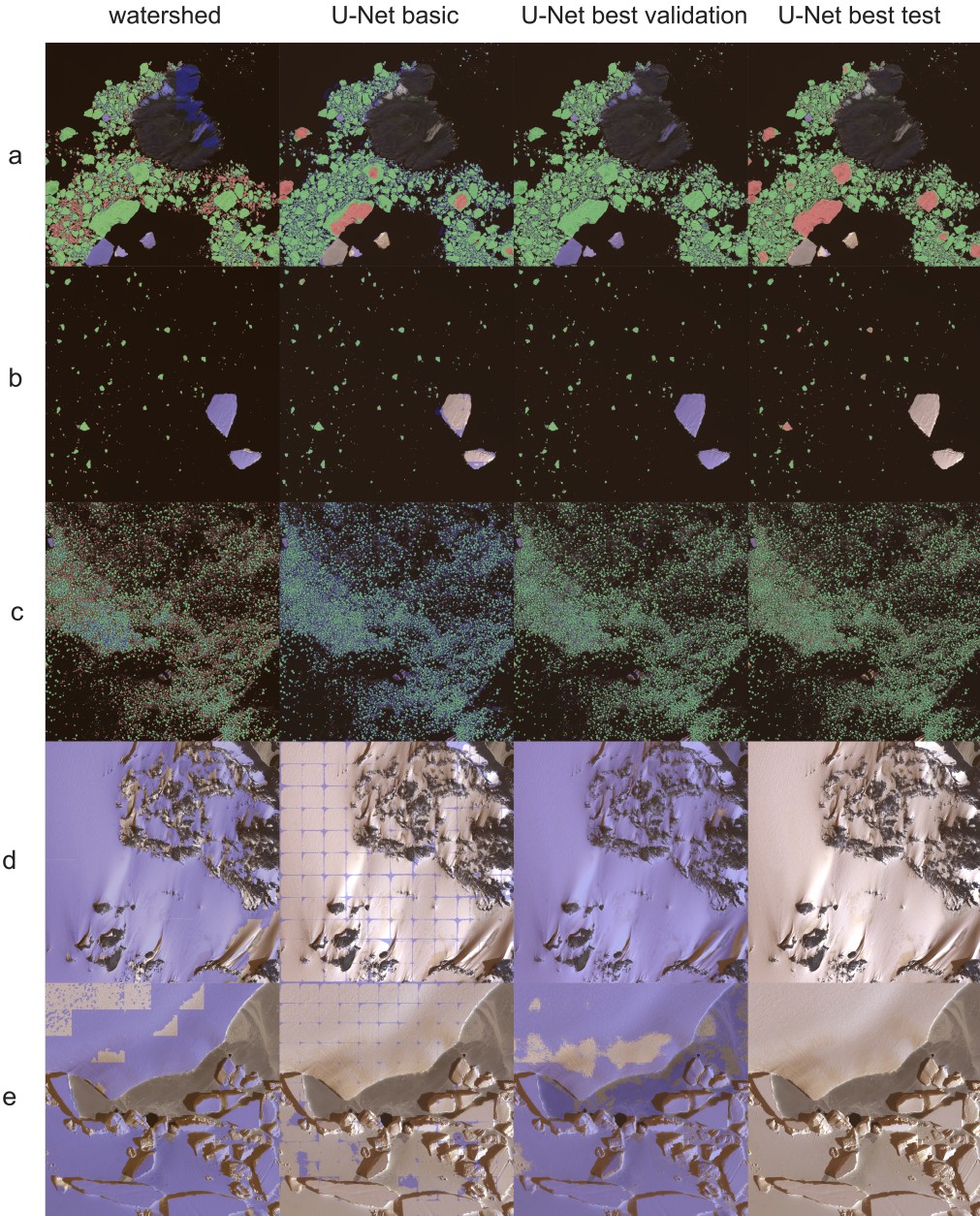

**Figure 5.** Output visualization. Model output at test scenes from 4 different sea ice extraction models, left to right: watershed segmentation, a basic U-Net, the best U-Net according to validation metrics, and the best U-Net according to test metrics. Test scenes are 3000 × 3000 m WV03 multispectral scenes from the Antarctic coastline tiled with a 50% overlap at the input size required by each model. Scenes (**a–c**) were chosen to illustrate model performance when ice floes are present, whereas scenes (**d,e**) illustrate the amount of false-positives generated by each model when predicting outside of pack-ice. True positives, false-positives and false-negatives shown in transparent green, purple and pink, respectively. Our final model generates few if any false-positive errors in land and fast-ice imagery, consistently avoids rock formations and icebergs, does not create artifacts at tiles edges, while capturing the majority of pack-ice within predicted masks. Imagery copyright Maxar, Inc., Westminster, CO, USA, 2021.

## 4. Discussion

### 4.1. Model Out-of-Sample Performance

Even with a modest-sized hand-labeled training set, our CNN-based method largely outperforms threshold based methods, represented here by a sequential watershed segmentation algorithm, quantitatively (Table 3) and qualitatively (Figure 5). Even after running a comprehensive hyperparameter search (Figure 4), experiments using direct outputs of watershed segmentation as weak-labels (i.e., training set = hand + watershed) underperformed those with hand-labeled data only (Table 3). This result is expected if we take into account situations where there is lighter and darker pack-ice within the same patch, in which case the watershed algorithm will only retrieve the lighter-colored floes (e.g., Figure 2, patches 2 and 4), creating misleading annotation masks. Using our synthetic image approach (Figure 2), however, adds valuable supervision to our semantic segmentation CNNs, improving test F1 score by a considerable margin (Table 3), but incurring more false-positive errors when predicting outside of pack-ice (Figure 5). With evaluation metrics to further penalize poor performance in background scenes and provide a better representation of true out-of-sample performance, we improved our test F1 score even further (Table 3), reaching over 0.75 in our comprehensive hand-labeled test set. Besides generating better prediction masks for ice floes, CNN based methods are particularly advantageous because they are able to understand context, and thus produce considerably fewer false-positives than threshold-based methods in at least three scenarios: (1) outside of pack-ice (Figure 5d,e); (2) in coastal areas; and (3) when icebergs are abundant. As one of our main goals was to evaluate CNNs as sea ice extraction tools, we did no post-processing on the output. There are several post-processing steps developed to improve the output from threshold-based or clustering-based methods that could also be beneficial if applied to our CNN-based pipeline (e.g., [16,17]), especially with obtaining better ice floe boundaries when floes are tied together [38].

### 4.2. Hyperparameter Search

Given the vast room for design choices with model architecture, loss functions, data augmentation routines and training schedule and recent breakthroughs in GPU-accelerated parallel computing, the hyperparameter search has become a key step in developing ML pipelines and an active research field (e.g., [39,40]). To allow an adequate exploration of design choices in a feasible time-frame, our hyperparameter search (Figure 4) focused on experimenting with input size, data augmentation routines, choice of loss function, choice of training set, ratio of negative to positive samples on training batches, and wether to fine-tune from a previous model. Surprisingly, with a few exceptions such as the underperforming LogCosh loss function [33] and the success of our mixture of Focal loss and Dice loss, there were no significant effects from our design choices in terms of validation F1 score (Figure 4, mid panel in the lower part of the figure). Some settings, in particular data augmentation, would merit more experimentation, both in terms of further exploring the transformations adopted in this study by experimenting with their hyperparameters, and experimenting with novel transformations (e.g., [41,42]). Another promising direction with further hyperparameter studies is testing larger input sizes, as there seems to be an increasing trend in the median validation F1 score as we increase input size (Figure 4, left panel in the upper part of the figure). We did not pursue that, however, because that would drastically reduce the size of our training batches since increasing input size has a quadratic effect on GPU memory usage. One design aspect that we did not touch in the present work and is particularly of interest to DL-based remote sensing applications is taking full advantage of multispectral bands. Apart from having a similar effect to GPU memory utilization as adding larger input sizes, taking 8-band images as inputs to our CNN model would require a series of modifications to the CNN architecture, making it less preferable than other important design choices included in our hyperparameter search when taking into account developer time allocation and computing resource utilization.

### 4.3. Fine-Tuning Experiments

Fine-tuning a model from previous model weights [31] obtained from training with a large, general purpose dataset like the Imagenet challenge dataset [22] has become a staple when training computer vision CNNs. Such an approach is grounded on the generality of low-level structures like edges and simple shapes across applications, and often focuses on re-training only the last few layers in the CNN [43], which focus on more high-level structures. Fine-tuning is especially useful when there is a scarcity of labeled data. Existing model weights, however, are largely based on natural images from frontal angle, hindering their usability aerial or satellite imagery based computer vision solutions, where the camera is always at an approximately 90°angle and the scale at which objects are presented is more or less fixed. Alternatively, fine-tuning for semantic segmentation models can be achieved by using patch-level labels to train a classifier model and swapping the weights from the original model backbone by the classifier parameter weights. Another approach is to fine-tune from a model trained at a different input size, aiming to be more scale-invariant. We experimented with both approaches, and failed to obtain any improvement when fine-tuning from a classification model, while obtaining some sparse improvements when fine-tuning from previous semantic segmentation models (best performing models in 3 out of 12 of our hyperparameter brackets used fine-tuning from previous models). Interestingly, the trend line for the effect of learning rate in validation F1 scores changes sign for fine-tuning experiments (Figure 4), potentially meaning that high learning rates could be breaking low-level feature representations from loaded model parameter weights. Since we decreased our learning rate during training whenever validation performance reached a plateau, results on the latter could have arisen by allowing the model to get out of a local minima, similar to a warm-restart learning rate scheduling policy [44].

### 4.4. Conclusions

Though sea ice models at course resolution following the plastic continuum approach [45] can generate sensible predictions of several key features (e.g., sea ice thickness, sea ice cover) and will remain useful for climate modelling [46], their assumptions do not hold at finer scale [47]. The added granularity provided by our solution allows a better treatment of important phenomena such as the formation of fractures and leads that can substantially alter the structure of sea ice as it allows more short-wavelength absorption by the ocean [48]. Additionally, since tasked high-resolution satellite imagery (e.g., WV-3) can be retrieved at specified locations within hours, our approach can enhance sea ice detection for shipping and logistics with a broader range of action than ship-based camera approaches (e.g., [24,25]). Because of its reliability outside of pack-ice areas (e.g., Figure 5), our pipeline is capable not only of producing sharp ice floe segmentation masks but detecting the presence of floes in very-high resolution imagery. Our fully automated, context-robust approach allows us to leverage modern GPUs to monitor fine-scale sea ice conditions at continental level. Finally, our semantic-segmentation approach could be expanded to segment and classify different fine structures in Antarctic and Arctic landscape provided we have plenty labeled images at a passable quality standard.

**Author Contributions:** Conceptualization, B.C.G.; methodology, B.C.G.; data annotation, B.C.G.; resources, H.J.L.; data curation, H.J.L.; writing—original draft preparation, B.C.G. and H.J.L.; writing—review and editing, B.C.G. and H.J.L.; funding acquisition, H.J.L. All authors have read and agreed to the published version of the manuscript.

**Funding:** This research was funded by the National Science Foundation (Award 1740595). Geospatial support for this work provided by the Polar Geospatial Center under NSF-OPP awards 1043681 and 1559691.

**Data Availability Statement:** Satellite imagery used for this analysis was provided by the Polar Geospatial Center under the purview of the NextView license and may be obtained from the original data vendor Maxar.

**Acknowledgments:** We acknowledge use of Stony Brook University's SeaWulf computing cluster and significant computational resources provided by the Institute for Advanced Computational Sciences.

**Conflicts of Interest:** The authors declare no conflict of interest. The funders had no role in the design of the study; in the collection, analyses, or interpretation of data; in the writing of the manuscript, or in the decision to publish the results.

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
