# Peer review of "Fine-Scale Sea Ice Segmentation for High-Resolution Satellite Imagery with Weakly-Supervised CNNs"

_remotesensing, doi:10.3390/rs13183562_

Round 1

Reviewer 1 Report

The work presents a variant of U-net architecture on Recent background to detect fine-scale sea ice. The proposed method was compared to the existing watershed algorithm for benchmarking. Watershed has its limitations and many other powerful methods are now available, maybe would be to compare with other state-of-the-art methods such as mask r-cnn.

Here are some of my comments

  1. There are types in spacing in lines 2, 27, and some others
  2. Github link of the code is not given though mentioned
  3. Could you please elaborate more on the augmentation and their rationale and impact on the performance?
  4. What is the reliability of the manual annotation? how consistent they are?
  5. Table 2 needs a bit more clarity on the models used in column 1.
  6. How did you decide on which loss function to use?

Overall, the paper is very well written and described.

Author Response

Reviewer 1

Comments to the Author

General Comments:

The work presents a variant of U-net architecture on Recent background to detect fine-scale sea ice. The proposed method was compared to the existing watershed algorithm for bench-marking. Watershed has its limitations and many other powerful methods are now available, maybe would be to compare with other state-of-the-art methods such as mask r-cnn.

REPLY: We thank the reviewer for the considerations and agree that the watershed algorithm may not be an ideal baseline for gauging our model performance. We chose not to go with Mask R-CNN or any other instance segmentation models, however, because they require individualized bounding boxes and masks which would be unfeasible in the context of ice floe segmentation. With that said, we do provide a naive U-Net implementation (table 3) as a serious contender for other implementations.

Specific comments:

  • There are types in spacing in lines 2, 27, and some others

REPLY: Typos were fixed, we thank the Reviewer for the careful inspection.

  • GitHub link of the code is not given though mentioned
    REPLY: Link to the Github code repository added in Line 224.

  • Could you please elaborate more on the augmentation and their rationale and impact on the performance?
    REPLY: We agree with the Reviewer that our explanation of our data augmentation pipeline lacks clarity, we added more details on the rationale for our choices in the Methods section and discussed the results under different data augmentation regimes briefly in the Discussion.

  • What is the reliability of the manual annotation? how consistent they are?
    REPLY: To minimize inconsistencies, our manual annotations were entirely done by the first author; the author revised the hand-labeled dataset several times to spot incongruences and often redid the annotation mask for a particular scene when it was found to be inconsistent with the dataset.

  • Table 2 needs a bit more clarity on the models used in column 1.
    REPLY: We added more details on each model in the caption for Table 2.

  • How did you decide on which loss function to use?
    REPLY: Our choice of loss function was done by starting with a broad set of plausible loss functions for semantic segmentation models, including novel solutions from the medical imaging literature, and letting the validation F1 score during hyperparameter tuning decide which loss function was the best fit for our use case. We added more explanation for our loss function choice rationale in the Methods section.

Reviewer 2 Report

Please see the pdf

Reviewer 3 Report

This paper presents an interesting exploration of using U-NET to extract sea ice from high-resolution Worldview-3 multispectral satellite imagery. I think that this paper is interesting and the topic is suitable to Remote Sensing. This paper is well written and structured, the methodologies are well described, and result properly discussed. The paper can be accepted for publication subject to minor revisions.

Comments:

1.Please add a brief description about the Worldview-3 (e.g., waveband setting,
acquisition time etc.) and specify the three bands used in the study since the WV-3 imager has 8 multispectral bands. If possible, the reviewer is also interesting in whether multispectral bands will help to improve the sea ice extraction accuracy.
2. It is better to give the architecture of the proposed U-Net model for sea ice extraction. Meanwhile, please add more details about the training procedure in Section 2.3, i.e., the batch size (Line 143-144, please specify the number of batch size), the number of training epochs, the train and validation learning curves.
3. Line 110-111, this paper removes objects that are too small to be a floe from the mask, and it can be seen in Figure 2 that small sea ice floes are masked out. Please specify the size threshold.
4. Please add more explanations and references about some terminologies. For example, F1 score in Line 171, IoU in Line 171. Meanwhile, Line 144, “f1-score” should be “F1 score”.
5. Line 130: “DICE coefficient” should be “Dice coefficient”?
6. Line 137, “All our CNN training experiments ran on PyTorch v1.8.0 in Python” should be “All our CNN training experiments were ran on PyTorch v1.8.0 in Python”.
7. Page 7, Table 2 Row 13, the patch size is 512 for U-Net with hand + watershed, not 384? Meanwhile, what does N stand for?
8. Figure 3, what does BCE stand for? DICE and Dice are the same? Please add more explanations.
